

# A hybrid GA-PSO strategy for computing task offloading towards MES scenarios

Wenzao Li[1,2], Xiulan Sun[1], Bing Wan[3], Hantao Liu[4], Jie Fang[1] and Zhan Wen[1]

[1] College of Communication Engineering, Chengdu University of Information Technology, Chengdu, China
[2] Network and Data Security Key Lab. of Sichuan Pro., University of Electronic Science and Technology of China, Chengdu, China
[3] School of Software, Chengdu Polytechnic, Chengdu, China
[4] Educational Informationization and Big Data Center, Education Department of Sichuan Province, Chengdu, China

Corresponding author
Xiulan Sun, sxlan12@163.com

## ABSTRACT

As a new type of computing paradigm closer to service terminals, mobile edge computing (MEC), can meet the requirements of computing-intensive and delay-sensitive applications. In addition, it can also reduce the burden on mobile terminals by offloading computing. Due to cost issues, results in the deployment density of mobile edge servers (MES) is restricted in real scenario, whereas the suitable MES should be chosen for better performance. Therefore, this article proposes a task offloading strategy under the sparse MES density deployment scenario. Commonly, mobile terminals may reach MES through varied access points (AP) based on multi-hop transmitting mode. The transmission delay and processing delay caused by the selection of AP and MES will affect the performance of MEC. For the purpose of reducing the transmission delay due to system load balancing and superfluous multi-hop, we formulated the multi-objective optimization problem. The optimization goals are the workload balancing of edge servers and the completion delay of all task offloading. We express the formulated system as an undirected and unweighted graph, and we propose a hybrid genetic particle swarm algorithm based on two-dimensional genes (GA-PSO). Simulation results show that the hybrid GA-PSO algorithm does not outperform state-of-the-art GA and NSA algorithms in obtaining all task offloading delays. However, the workload by standard deviation approach is about 90% lower than that of the GA and NSA algorithms, which effectively optimizes the performance of load balancing and verifies the effectiveness of the proposed algorithm.

## INTRODUCTION

### Background and motivation

With the advent of 5G and the Internet of Things, the demand for mobile applications is increasing. Scholars believe that mobile edge computing (MEC) is an emerging technology to meet the needs of mobile network business (*Wang, Cheng & Chen, 2020b*). A mobile edge network is commonly considered as a three-tier architecture consisting of core layer,

edge layer, and user layer. The MEC network architecture assists computing and data storage resources near the terminal equipment. These solutions can effectively reduce computing latency through cloud computing, thereby alleviating network congestion. This solution has become a hot topic due to its excellent delay performance and security characteristics (*Chen et al., 2021*). Mobile devices with limited resources can obtain excellent performance (*Mahmud, Ramamohanarao & Buyya, 2018*; *Mao et al., 2017*) and perform tasks efficiently by offloading computing tasks to nearby mobile edge servers. Meanwhile, energy consumption of mobile devices can be reduced (*Du et al., 2020*) and battery life of devices can be extended (*Feng et al., 2019*).

Edge server (ES) is a new mobile edge computing framework (*Zhao et al., 2018*), which has computing and data storage capabilities. The MEC network architecture can improve the efficiency of real-time data analysis and processing by placing edge servers in network base stations or access points (*Xu et al., 2020*; *Deng et al., 2021*). In actual application, numerous traditional task offloading strategies disregard some critical problems that need additional discussion. There are terminals with computing requirements that are typically distributed over a relatively wide area. Then, the geographic location of offloading tasks usually lead the mobile edge servers (MES) overload in some ultra-dense networks (UDN). Overmuch tasks are offloaded to the same edge server, which can lead to server overload and congestion, which not only affects the system performance and server lifespan, but also leads to a sharp drop in quality of service and quality of experience (*Guo, Liu & Zhang, 2018*; *He et al., 2019*; *Fan & Ansari, 2018*). Thus, it is vital to resolve the problem of load balancing among edge servers.

## Limitations of prior work

The optimization of two main indicators in the research of task offloading strategies has attracted great attention, namely, minimizing energy consumption and minimizing delay. There are existing works on optimizing these two indicators independently or simultaneously. In *Ding et al. (2019)* and *Pan et al. (2018)*, the propagation time and energy consumption of the MEC system were optimized using geometric programming and successive convex approximation, respectively. In multi-server and multi-task scenarios, *Zhang et al. (2021)* designed an optimization problem with the goal of minimizing the completion time of all tasks. *Wang, Xing & Xu (2020a)* proposed a wirelessly powered multi-user mobile edge computing system environment to minimize the total system energy consumption within a limited time horizon. *Fan et al. (2020)* aim to decrease the latency and energy consumption of mobile terminals during task processing. The research introduces a balance factor to flexibly adjust the minimum value between the energy consumption of the mobile terminal and the processing delay of the task (*Fan et al., 2020*). The efforts focus on addressing system energy consumption and computation latency in these artilces, while they have ignored the impact and consequences of unbalanced workload on mobile edge servers.

*Yu, Tang & Li (2019)* and *Lu et al. (2020)* proposed task offloading strategies based on reinforcement learning and deep reinforcement learning (DRL), respectively. Their researches has achieved the purpose of optimizing load balancing. *Mogi, Nakayama &*

*Asaka (2018)* mainly research load balancing among mobile edge servers when load conditions fluctuate dynamically. They proposed a load balancing approach, which is mainly used in IoT sensor systems. In the edge computing network of the joint cloud data center, *Dong et al. (2019)* proposed a task offloading algorithm, in order to solve the server selection problem of task offloading. This algorithm combines the advantages of task clustering method and firefly swarm optimization algorithm. *Guo et al. (2018)* proposed an efficient suboptimal algorithm by minimizing the energy consumption of each terminal through joint optimization. *Zeng & Fodor (2019)* transformed network wide resource allocation into a convex optimization problem to allocate communication and computing resources for users. *Tran & Pompili (2018)* decompose the optimization problem that minimizes task execution delays and user energy consumption into task allocation problem and resource allocation problem, and apply multiple algorithms to solve the problem. *Wang et al. (2019)* considered the limited power of equipment in the three-layer collaborative computing network and optimized the minimum average task duration. In the part of the literature, scholars considered the objective of load balancing optimization. However, the network scenarios considered by the scholars all have sufficient edge server resources and do not take into account the limited environment of edge servers. While in these research which considers the limited computing resources of edge servers or terminal devices, the optimization objectives of these studies are still focused on system energy consumption or offloading latency (*Zeng & Fodor, 2019*; *Tran & Pompili, 2018*; *Wang et al., 2019*).

## Challenges and solutions

A wireless metropolitan area network (WMAN) is composed of a large number of wireless access points (APs) (*Xu et al., 2015*; *Zeng et al., 2018*). Under the condition of limited deployment of edge servers, that is, when edge servers are deployed in some APs, the advantages of multi-hop edge computing networks can be used for a large number of terminals render computing services (*Al-Abiad, Zoheb & Hossain, 2021*). Yet, in such this scenario, there are numerous challenges to be resolved. The task offloading strategy determines which MEC server to process the offloading requirement. But because of the huge number of tasks and data received by APs, different task offloading strategies considerably alter the load balancing among edge servers. To achieve load balancing among servers, it is necessary to design a suitable global offload scheme for all tasks. In addition, APs receive offloaded large-scale tasks and transmit them to nearby MEC servers or other APs by utilizing wireless links for data delivery. However, multi-hop communication in the network may cause additional delays, so this article focuses on the strategy design for delay reduction for system performance.

## Contributions and organization

In this article, we consider task offloading in scenarios with limited deployment of edge servers in a wireless metropolitan area network. We propose an optimization problem to

jointly optimize the load balancing between edge servers and task offloading latency. The main contributions of this article are summarized as follows:

1. The scenario of deploying edge servers on some APs in the wireless metropolitan area network is reconstructed into a simple and easy-to-understand un-directed unweighted graph, and a multi-objective optimization problem for optimizing edge server load balancing and offloading latency are constructed. And we prove that the optimization problem is NP-hard.

2. The system is expressed as an un-directed and un-weighted graph, and a hybrid genetic particle swarm algorithm based on two-dimensional particles is proposed. The algorithm utilizes two-dimensional particles to represent the offloading decision, through the task grouping under the AP service scope, server selection, and path selection to achieve the optimization objective function.

3. The algorithm is simulated using the actual base station geographic location data set. Experimental results show that the proposed algorithm has a good effect on solving optimization problems.

The rest of this article is organized as follows. The second part elaborates on the system model. "Problem definition and proof" is the problem formation and related proof process. Detailed solutions are provided in "Solving method". "Simulation analysis" evaluates the performance of the proposed algorithm based on the simulation results. "Conclusion" concludes this article and future work.

## SYSTEM MODEL

Under the specific scenarios in this section, we build network models to derive optimized models for offloading time and load balancing in edge computing scenarios. Table 1 gives the main symbols and their meanings.

### Network model

As shown in Fig. 1, we include a multi-hop mobile edge computing scenario consisting of several APs and multiple servers. This article uses a connected un-directed graph $G = (A \cap S, E)$ to represent the network, A represents the set of wireless APs, $A = \{a_1, a_2, \ldots, a_n\}$, E represents the set of links between APs; when APs $a_i(i \in A)$ and $a_j(j \in A)$ links are connected, there is an edge $(i, j) \in E$; S represents the set of edge servers, $S = \{s_1, s_1, \ldots, s_m\}$. Where $n$ and $m$ respectively represent the number of APs and edge servers, and the value of $m$ must be much smaller than the value of $n$. Some APs in the network are deployed with edge servers with the same capacity. If an AP is deployed with an edge server, the AP and the server are collectively referred to as edge computing nodes. The user terminal under the radius $R$ of each AP service area has a task request offloading, in which the task request under the AP $a_i$ service range is represented as $T_i = \{t_1, t_1, \ldots, t_v\}$ (*Xiulan Sun & Li, 2022*). Each task is indivisible, and $d_v$ represents the data size of task $t_v$, $v_i$ represents the number of all tasks of $a_i$. $h_{ij}$ represents the number of tasks that $a_i$ offloads to the server $s_j$. In this article, each AP is connected with each other that are geographically close to each other through a wireless link, and the distance

**Table 1 Notations.**

| Notation | Definition |
|---|---|
| $G$ | Mobile edge computing network |
| $E$ | The set of links between base stations in the network |
| $B$ | The set of all base stations in the network |
| $S$ | The set of all edge servers |
| $R$ | Circle radius of AP's service area |
| $K_j$ | The collection of tasks computed by the server $s_j$ |
| $n$ | The number of APs |
| $m$ | The number of edge servers |
| $v$ | Total number of tasks received by all APs |
| $k_j$ | The number of tasks computed by the server $s_j$ |
| $T_i$ | The set of task sizes received on AP $a_i$ |
| $d_v$ | The data size of task $t_v$ |
| $h_{ij}$ | The number of tasks that AP $a_i$ offloads to edge server $s_j$ |
| $\alpha$ | Data transmission rate between APs |
| $\alpha'$ | The amount of data transmitted by the wireless link in one time slot |
| $\varepsilon$ | Time slot |
| $h^n$ | The number of hops in the transmission path |
| $\beta$ | Workload weight |

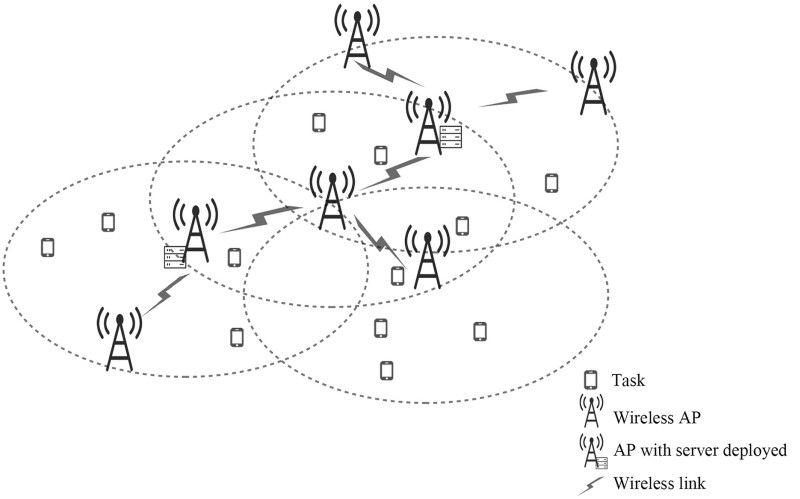

**Figure 1 The scenario with limited edge server deployment.**

between the two APs is less than $R$ before they can communicate. At the same time, the number of APs that each AP can communicate with each other does not exceed 3, and $\alpha$ represents the rate of data transmission between APs. In intelligent edge computing, after a large number of computing tasks are offloaded from mobile devices to nearby base stations, edge computing nodes need to determine how to allocate computing resources for

execution (*Xu et al., 2019*). The network's intelligent manager receives the servers' load information from the edge computing nodes and formulates the offloading strategy.

## Computation model

### Calculation of load balancing

Load balancing of MES is one of the main research issues in edge computing. Due to the uneven distribution of intensive tasks and edge servers, some edge servers may be overloaded, causing network congestion. An important purpose of researching load balancing of MES is to help improve resource usage, to assure that no single node is overloaded, to decrease mobile users' waiting time, and to improve mobile users' experience (*Xu et al., 2019*).

This article does not consider the process of task uploading, because although a communication model is introduced in most research work, the allocated channel bandwidth, uploading power and signal interference of each mobile terminal are considered to be fixed, which will lead to the fixed task uploading delay, such as references (*Chen, Zhou & Xu, 2018*; *Wang et al., 2019*). If parameters such as channel bandwidth allocation are regarded as dynamic variables, the entire system model will be too complex to be resolved. Thus, by researching the literature (*Mondal, Das & Wong, 2020*; *He et al., 2019*; *Dong et al., 2019*), this article mainly studies the workload balance of edge servers in mobile edge computing, without considering the communication model of task uploading.

By referring to the research literature, this article mainly studies the computing load balancing of edge servers in mobile edge computing, mainly balancing the task load that has reached the AP but has not started to execute, and does not consider the communication model of task uploading.

This research uses standard deviation to evaluate the workload balancing of edge servers. We know that $m$ edge servers are located among APs, then we calculate the workload of each edge server $s_j$ as $w_j$ and the workload of server $s_j$ as:

$$w_j = \sum_{k=1}^{k} d_k \tag{1}$$

where $k_j$ represents the number of tasks calculated by server $s_j$, $K_j$ represents the task set calculated by server $s_j$. The average workload of all edge servers is expressed as follows:

$$w_{ave} = \frac{1}{m} \sum_{j=1}^{m} w_j \tag{2}$$

The standard deviation of the workload can be calculated as follows:

$$w_B = \sqrt{\frac{\sum_{i=1}^{m} (w_i - w_{ave})^2}{m}} \tag{3}$$

It is straightforward to know that the smaller the value of the standard deviation, the more balanced the workload of each edge server (*Xiulan Sun & Li, 2022*).

### Calculation of offloading delay

The above mode ignores the time consumption of task upload process, accordingly we principally consider the time when the task is offloaded to the target server through the AP. The tasks on each AP are offloaded to the edge server for computing through a wireless link. If the AP is deployed with an edge server, we assume that the received tasks on the AP will have the smallest transmission delay, which can be ignored. If the AP is not deployed nearby an edge server, then the received task of the AP needs to be forwarded to the nearby edge server through the linked adjacent base station for computing, and the cumulative delay of the multi-hop transmission of each task constitutes the task offloading delay (*Xiulan Sun & Li, 2022*).

Task data in the network is transmitted over the wireless link in parallel. In this article, a time slot $\varepsilon$ is defined as 1 s, then the value of the data amount $\alpha'$ transmitted in a time slot is equal to $\alpha$. In a certain time slot, if the sum of the data amounts of b tasks under the service range of an AP is less than or equal to $\alpha'$, and the sum of the data amounts of $b + 1$ tasks is greater than $\alpha'$. The current time slot only transmits the data amount of $\alpha'$, and the data that has not been transmitted in the $(b + 1)$th task needs to be transmitted in the next time slot. Until there are no outstanding task requests under the service scope of all APs in the network.

The delay for all tasks to be completely offloaded in the network is

$$t_{all}^{tran} = max\{t_1^{tran}, t_2^{tran}, \ldots, t_n^{tran}\} \tag{4}$$

$t_{all}^{tran}$ indicates the time required to complete the offloading of all user tasks under the AP service scope with serial number n.

## PROBLEM DEFINITION AND PROOF

### Problem formulation

In our research, we focus on reducing the workload standard deviation among edge servers, while minimizing the latency for all task offloading to complete. Then our objective function is formulated as follows:

$$P1 : Minimize\left[w_B, t_{all}^{tran}\right]$$

$$s.t. \quad a : \sum_{j=1}^{m} h_{ij} = v_i \ i \in A, j \in S$$

$$b : \sum_{j=1}^{m} k_j = \sum_{i=1}^{n} v_i \ i \in A, j \in S \tag{5}$$

$$c : \sum_{i=1}^{b} d_i = \alpha' \ i \in A$$

Constraint a ensures that all task requests under the AP $a_i$ service scope will be offloaded to the edge server for processing, and no tasks have not been offloaded. Constraint b ensures that all tasks in the network are offloaded to the server for computation. Constraint

c means that the amount of task data transmitted per second by the wireless link is less than or equal to $\alpha'$.

## NP-hard proof

This subsection will prove that the proposed offload optimization problem is an NP-hard problem. We can summarize the offload optimization problem as follows: Consider the task offloading problem in a given un-directed complete graph $G' = (A, S')$, where $A$ is the position of each AP $a_i \in A$ and $S'$ $S_j \in S$ is the position of each MES. The task offloading problem is to offload the tasks within the service range of each AP into MES for calculation. The optimization goal is to reduce the standard deviation of workload among MES and minimize the delay of all task offloading completion.

In mobile edge computing environment network $G = (B, S)$, a task offloading problem whose optimization goal is MES load balancing has been proved to be an NP-hard problem (*Chen et al., 2021*). $B$ stands for small base station (SBSs). We construct a mobile edge computing network $G' = (A, S')$ from network $G = (B, S)$, where $A = B$, $S' = S$. Meanwhile, in $G'$, we consider the case where the number of deployed MES is less than the number of APs, and simultaneously optimize the standard deviation of the workload and the delay of task offloading completion. Therefore, the optimal solution of the proposed offloading problem is also the optimal solution of $G$. Since the task offloading problem in $G$ is an NP-hard problem, our proposed offloading problem is also an NP-hard problem.

## SOLVING METHOD

In the previous section we proved that the optimization problem of Eq. (5) is NP-hard. Currently, in the study of MEC systems, researchers design MEC offloading strategies mainly in the following ways: (1) Designing the offloading strategy based on convex optimization theory. This approach may take a long time. However, the original intention of mobile edge computing is to shorten the computation delay, and this approach goes against this. (2) Using artificial intelligence technology to design offloading strategy. But reinforcement learning, machine learning, and other techniques require large amounts of historical data for training and learning. Moreover, due to the complexity and dynamics of the MEC research environment, the authenticity of the training data is highly desired. If the correlation between the training data and the real-time data is low, the offloading strategy devised by the AI technique cannot achieve the optimization purpose. (3) Designing the offloading strategy using relevant heuristic algorithms. The offloading decisions designed in this approach have low complexity and great applicability. Therefore, in this article, we propose an efficient algorithm to solve this problem by innovating heuristics for the purpose of this study.

## Details of hybrid genetic particle swarm optimization based on two-dimensional particles

Genetic algorithms and particle swarm algorithms are frequently used in searching for optimal offloading strategy, and both have their own characteristics (*You & Tang, 2021*; *Liu & Zhang, 2019*). Because of its population diversity, the genetic algorithm is suitable

for global search. Nevertheless, due to the blindness of genetic crossover, mutation and other operations, the convergence time is long. The particles in the particle swarm algorithm have memory, and rapid convergence can be achieved by adjusting the speed and position of the particles. However, the population diversity and search range of the particle swarm optimization algorithm are limited and it is easy to fall into the local optimal solution (*Zewei et al., 2021*). Consequently, based on the model of this article, combined with the basic idea of genetic algorithm and particle swarm optimization, a hybrid genetic particle swarm algorithm is proposed. This algorithm combines the advantages of particle swarm optimization and genetic algorithm, improves the diversity of the population and the global search ability, and avoids the algorithm from falling into the local optimal solution.

Particle position vector encoding: In this article, the particle is defined as a two-dimensional array, the number of rows is $n$, the number of columns is $m$, and $X = [x_1, x_2, \ldots, x_n]$. $x_n$ is an array in the nth row, representing the task offloading scheme on AP $a_n$. The elements in the array are the numbers in 0 to $m$: 0 is used to represent the placeholder. If the non-zero number in the array has $a$ bit, all task requests are divided into $a$ shares. A non-zero number indicates that tasks are offloaded to the edge server corresponding to a non-zero number. If the number of APs on the network is 10, the number of sequences of APs ranges from 1 to 10. If the number of MES is 5 and APs with sequence numbers 1, 2, 3, 4, 5 deploy MESs, then the numbers in the particle position vector can only be 0, 1, 2, 3, 4, 5. The particle position vector $X$, which should be a $10 \times 5$ array depending on the number of APs and MESs, represents the offloading decision for the whole system. The one-dimensional array $x_1$ in the first row represents the offloading scheme for the tasks received by $a_1$. When $x_1 = [0, 1, 5, 3, 0]$, there are three nonzero numbers, which means that all task requests in the service range of $a_1$ are divided into three parts after sorting according to the size of data. The tasks of these three parts are offloaded to MESs deployed by $a_1$, $a_5$, and $a_3$ in order.

Particle velocity vector encoding: Velocity represents the span of offloading tasks to other servers, denoted by $V = [v_1, v_2, \ldots, v_n]$, the number of rows and columns is either $n$, $m$. $v_n$ is the array of the nth row, represents a change in the task offloading scheme of the AP $a_n$ service scope. If the position vector is $x_1$, the particle velocity vector $v_1$ is $[1, 2, -1, 0, 1]$. Thus the particle position vector $x_1$ is updated to $[1, 3, 4, 3, 1]$. The change of the position vector means that task requests under the service scope of the AP with sequence number one are split into five groups, and the task requests divided into five groups are offloaded to the edge servers with sequence numbers 1, 3, 4, 3, and 1, respectively.

Fitness function: For each particle, the fitness value depicts the quality of the task offloading decision expressed by the particle, utilizing Eq. (6) as the fitness function $f$. The smaller the fitness value, the better the fitness of the particle. Therefore, our goal is to obtain the particle with the smallest fitness value during the iteration of the algorithm.

$$f = \beta * \frac{w_B}{w_{B(max)}} + (1 - \beta) * \frac{t_{all}^{tran}}{t_{all(max)}^{tran}} \tag{6}$$

In Eq. (6) $\beta$ and $(1 - \beta)$ represent the weight of workload standard deviation and task offloading delay, respectively. In this article, let $w_{B(max)}$ and $t_{all(max)}^{tran}$ be the maximum $w_B$ and $t_{all}^{tran}$ obtained at the initial iteration of the algorithm.

In the similar problem of shortest path, the Dijkstra algorithm, Floyd algorithm, A* algorithm and other algorithms are commonly used to solve the problem. However, the time complexity of the single iteration path search process of Floyd algorithm is higher than that of Dijkstra algorithm, and the algorithm speed is slower (*Li, Tong & Zhang, 2022*). The Bellman-Ford algorithm has the problems of extreme redundancy and low efficiency (*Wang, 2018*). The A* algorithm can be regarded as an extension of Dijkstra's algorithm, but the heuristic function in the algorithm will affect the behavior of the A* algorithm and may cause the A* algorithm to slow down. However, using the Dijkstra algorithm to solve the shortest path correlation problem, the path search procedure is computationally efficient with low time cost. Dijkstra's algorithm is based on graphs to solve the difficulty of the shortest path and generate the shortest path tree. This algorithm is generally used in the study of path planning problems (*Wang et al., 2022*; *Sun, Fang & Su, 2021*). Thus, when the task is offloaded from the initial AP to the destination edge server, our algorithm obtains the offloading path of the task *via* the Dijkstra algorithm.

In this article, the data transmission rate between APs is made the same, so the network can be expressed as an un-directed and unweighted graph. When the element in the array of the particle swarm is a number in 1 to $m$, it indicates which server to offload to. The input of the Dijkstra algorithm is the adjacency array of the un-directed unweighted graph, the sequence number corresponding to the initial AP and the destination server. And the output is the shortest path between the initial node and the server node.

The process of the hybrid genetic particle swarm optimization algorithm based on two-dimensional particles is as follows:

(1) Initialize the particle swarm according to the parameter constraints, and obtain the initial velocity and initial position of each particle.

(2) Calculate the offloading delay and workload standard deviation through the offloading scheme represented by the particle position vector. When the task starts offloading, the task information of each AP and edge server is updated every time slot until there are no task requests under the service scope of all APs.

(3) Calculate the fitness value of each particle, save or update the historical best fitness value *Pbest* of each particle.

(4) If the minimum fitness value in the particle swarm is smaller than the group's historical optimal fitness value, update the group's historical optimal fitness value *Gbest*.

(5) Sort the particle swarm according to the order of fitness value from small to large. Join the elite retention strategy to directly conserve the speed and position of the top 10%

particles after sorting. At the same time, the speed and position of the top 90% of the sorted particles are updated and saved after the update.

(6)  Equations (7) and (8) are used to update the velocity and position of the particle.

$$v_i^{h+1} = wv_i^h + c_1 r_1 (Pbest^h - x_i^h) \tag{7}$$
$$x_i^{h+1} = x_i^h + v_i^{h+1} \tag{8}$$

In Eqs. (7) and (8), the superscript $h$, $h+1$ represents the number of iterations, and $w$ is the inertia weight that regulates the search for space exploration. $c_1$ and $c_2$ are the self-perception factor and the overall perception factor, respectively, with a value of 2. $r_1$ and $r_2$ are two random numbers in the range $(0, 1)$. When updating the speed $v_i^{h+1}$, when the obtained value is not an integer, only the integer part is retained. And judge whether it exceeds the maximum speed $v_{max}$ and the minimum speed $v_{min}$, if not within the range, then regenerate. After the position is updated, it is necessary to judge whether the value of each digit in the position is between 0 and $m$. If the value is negative, take its opposite; if the value is greater than $m$, subtract $m$ from the value.

(7)  When the number of iterations is less than or equal to the maximum number of iterations, repeat steps (2) (3) (4) (5) (6); otherwise the iteration terminates.

The inertia weight $w$ in the algorithm, this article adopts the method of linear decline in $w$ through Eq. (9) (Yi & Zijiang, 2020). The particle swarm explores the large area at the beginning of the iteration and the approximate position of the optimal solution at the later stage of the iteration through the weight change of linear descent. In the process of weakening the inertia weight, the particle velocity is reduced, and the precise local search is started (Dongqiang & Xiaoxia, 2017).

$$w^h = (w_{ini} - w_{end}) * \frac{(iter_{max} - h)}{iter_{max}} + w_{end} \tag{9}$$

In Eq. (9) $iter_{max}$ is the maximum number of iterations, $h$ is the current number of iterations, $w_{ini}$ is the initial inertia weight, and $w_{end}$ is the inertia weight when the iteration reaches the maximum number of iterations.

## Algorithmic time complexity analysis

The proposed algorithm is mainly based on the GA and PSO algorithmic innovations. In algorithm, the number of iterations of the particle swarm is $i$, the size of the particle population is $p$, and the shape of the particle position vector is $n$ rows and $m$ columns. In the iterative update of the particle swarm, each iteration mainly computes the fitness value and updates the particle position vector. The calculation of fitness value and updating of particle position vector are related to the shape of particle position vector, and the time complexity of both parts is $O(pmn)$. Then, the entire iteration time is approximately $O(2ipmn)$. Therefore, the time complexity of genetic algorithm is $O(ipmn)$.

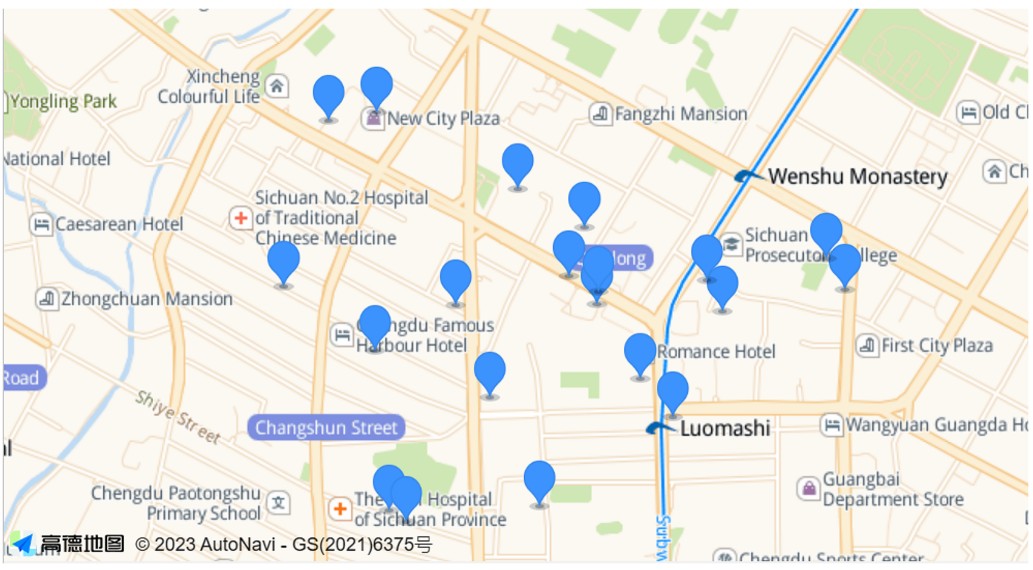

**Figure 2** **The location map of some real base stations in Jinniu District, Chengdu.**

## SIMULATION ANALYSIS

This section illustrates how to conduct simulation experiments to determine the effectiveness of our solution, which is simulated using Python.

### Simulation parameters

In the simulations presented in this article, we selected a certain range of real-world base station locations in Jinniu District of Chengdu City for our experiments. As shown in the Fig. 2. According to the proportion of the area range, we finally determined the location of 20 base stations (AP). The coverage radius of each base station (AP) was 750 m (*Al-Abiad, Zoheb & Hossain, 2021*). In the experiments, we validate the effectiveness of the proposed algorithm under different total number of tasks and make comparisons. The total number of tasks on the network is 3,000, 4,000, 5,000, 6,000, and 7,000. The data size of each task is [7,40]Mbit (*You & Tang, 2021*), and the data transfer rate between APs is 20 MB/s (*Fan et al., 2017*). At the same time, to balance the importance of the workload standard deviation with the task offloading delay of the edge server, we set the weight $\beta$ to 0.5. In the process of task offloading, we determined the unit of time slot as seconds (*Tang et al., 2021*; *Liao et al., 2021*). To obtain additional accurate results, we finally settled on a time slot $\varepsilon$ of 1 s. In algorithm, $w_{ini}$ and $w_{end}$ are determined to be 0.9 and 0.4, respectively (*Yi & Zijiang, 2020*). The parameter settings stated in this experiment are shown in Table 2.

In a network scenario with restricted server deployment, the number of servers is less than the number of APs, and some APs will deploy servers. This article uses the actual latitude and longitude of some base stations in Jinniu District, Chengdu, and converts them into two-dimensional coordinates. Then use the bisection K-means algorithm to split numerous AP points into $m$ clusters, and the value of $m$ is equal to the number of servers. The bipartite K-means algorithm overcomes the problem that the K-means algorithm is
| Parameter | Value |
|---|---|
| $n$ | 20 |
| $m$ | 4,5,6 |
| $v$ | 3,000, 4,000, 5,000, 6,000, and 7,000 |
| $R$ | 750 $m$ |
| $d_i$ | $[7, 40]Mbit$ |
| $\alpha$ | 20 M/s |
| $\beta$ | 0.5 |
| $\varepsilon$ | 1 s |
| $w_{ini}$ | 0.9 |
| $w_{end}$ | 0.4 |

**Table 2 Parameter settings.**

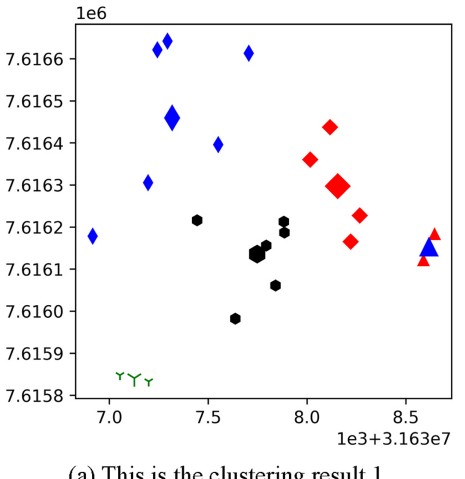

(a) This is the clustering result 1.

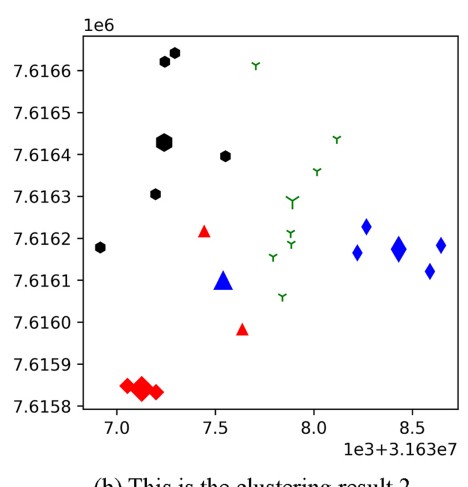

(b) This is the clustering result 2.

**Figure 3 These are the two clustering results from bipartite K-means.** (A) Clustering result 1; (B) clustering result 2.

sensitive to the initial cluster centroids. After clustering, we find the AP closest to the cluster center in each cluster, and set this AP as the AP where the server is deployed. After the location of the server is determined, each AP node communicates with up to three nodes. At the same time, the distance between the two APs is less than $R$ to communicate with each other, and the topology map of the two scenarios is randomly generated.

Figure 3 shows the result of the AP location after the bipartite K-means clustering algorithm. Points with the same mark belong to the same cluster, and the greater mark in the cluster represents the cluster center. Figure 4 is a corresponding network topology diagram generated randomly after selecting an AP node to deploy a server according to the clustering result. The small dots in each picture represent APs, the numbers on them represent the serial numbers of APs, and the triangle-shaped dots represent APs with deployed servers. Figures 3 and 4 show the topological scenario obtained when the number of deployed edge servers is 5. At the same time, we also obtain topological scenarios when

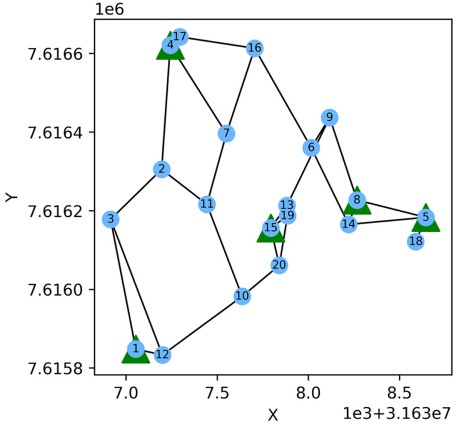

(a) The network topology map 1 corresponding to clustering result 1.

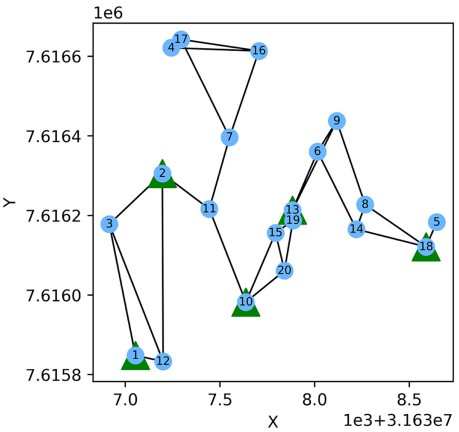

(b) The network topology map 2 corresponding to clustering result 2.

**Figure 4** **These are the two clustering results from bipartite K-means.** (A) Network topology map 1 corresponding to clustering result 1; (B) network topology map 2 corresponding to clustering result 2.

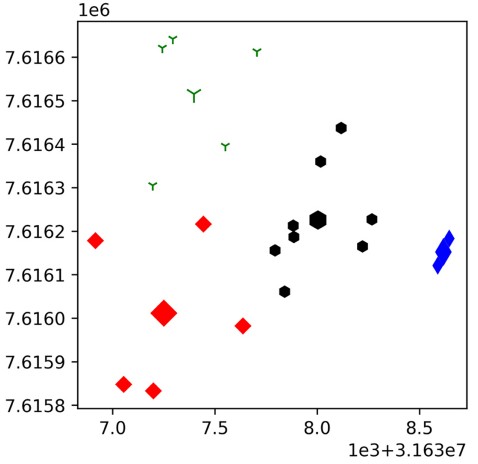

(a) This is the clustering result when the num -ber of deployed servers is 4.

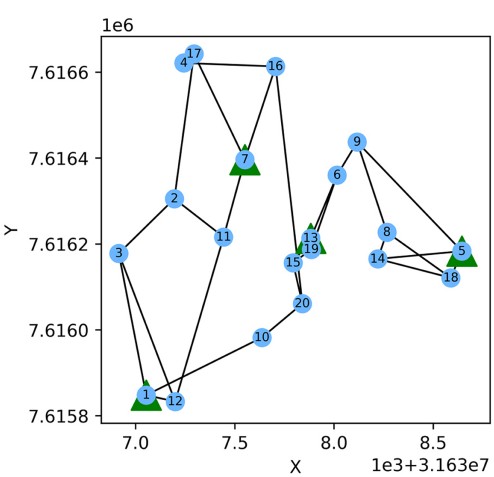

(b) This is the network topology when the number of deployed servers is 4.

**Figure 5** **This is the (A) cluster result and the (B) network topology when the number of deployed servers is 4.**

the number of deployed edge servers is 4 and 6. Figure 5 shows the clustering results and the network topology graph when the number of deployed edge servers is four. Figure 6 shows the clustering results and the network topology graph when the number of deployed edge servers is 6.

In simulation experiments, we compared the performance of several different offloading decisions in terms of workload balancing and offloading completion delay: Genetic algorithm (GA) based on one-dimensional genes (*Xiulan Sun & Li, 2022*), hybrid genetic particle swarm optimization based on two-dimensional particles (GA-PSO), and the nearest selection algorithm (NSA). The main idea of the offloading strategy of the nearest

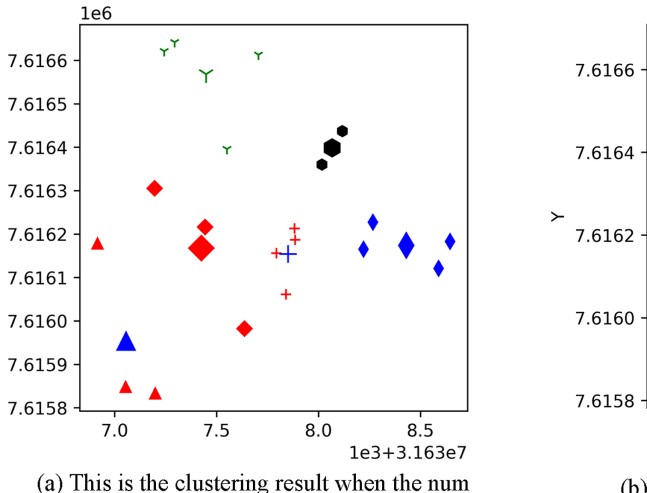 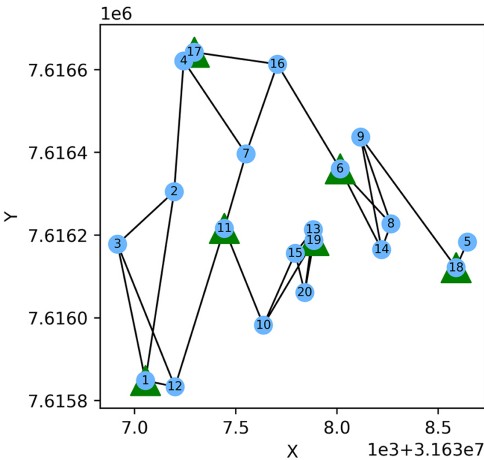

(a) This is the clustering result when the num
-ber of deployed servers is 6.

(b) This is the network topology when the
number of deployed servers is 6.

**Figure 6** This is the (A) cluster result and the (B) network topology when the number of deployed servers is 6.

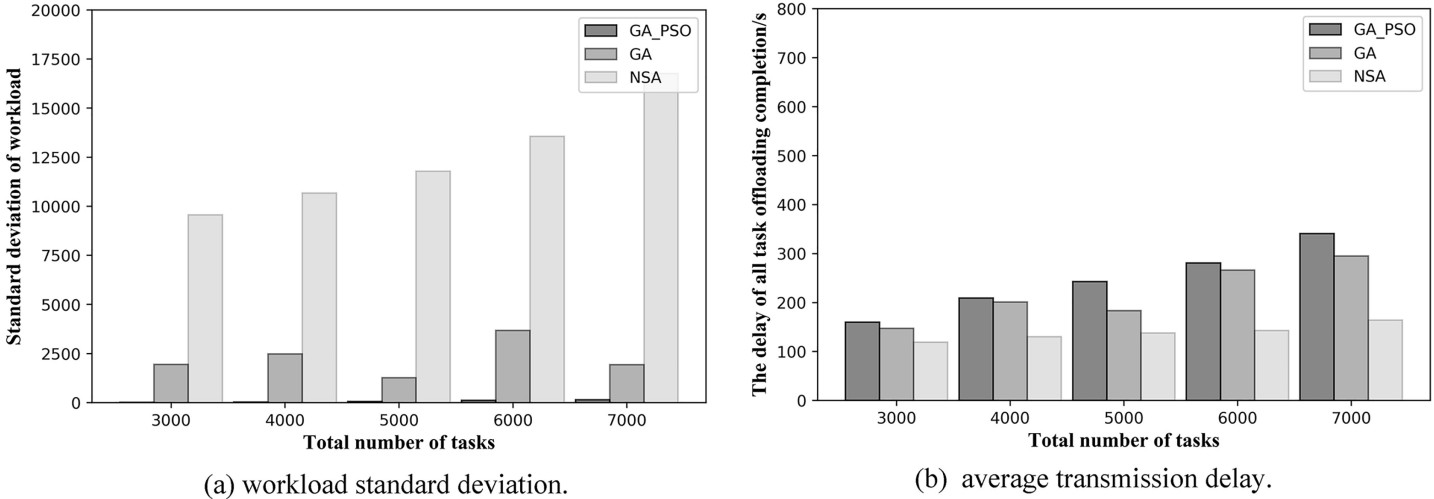

(a) workload standard deviation.

(b) average transmission delay.

**Figure 7** (A) Workload standard deviation and (B) average transmission delay of various algorithm offloading strategies in network topology scenario 1.

selection algorithm: according to the Dijkstra algorithm, each AP finds the server with the least number of hops in the offloading process, and all task requests under the service scope of each AP are offloaded to this server.

From Figs. 7 and 8, we can see that our proposed method outperforms the NSA and GA methods in general. Figures 7A and 7B respectively, show the results of load balancing and offloading completion delay when the total number of tasks in the network is different in topology scenario 1. For load balancing, it can be seen in Fig. 7A that the standard deviation of workload of GA-PSO algorithm is the lowest, followed by that of GA algorithm, while that of NSA algorithm is the highest. The lower the workload standard

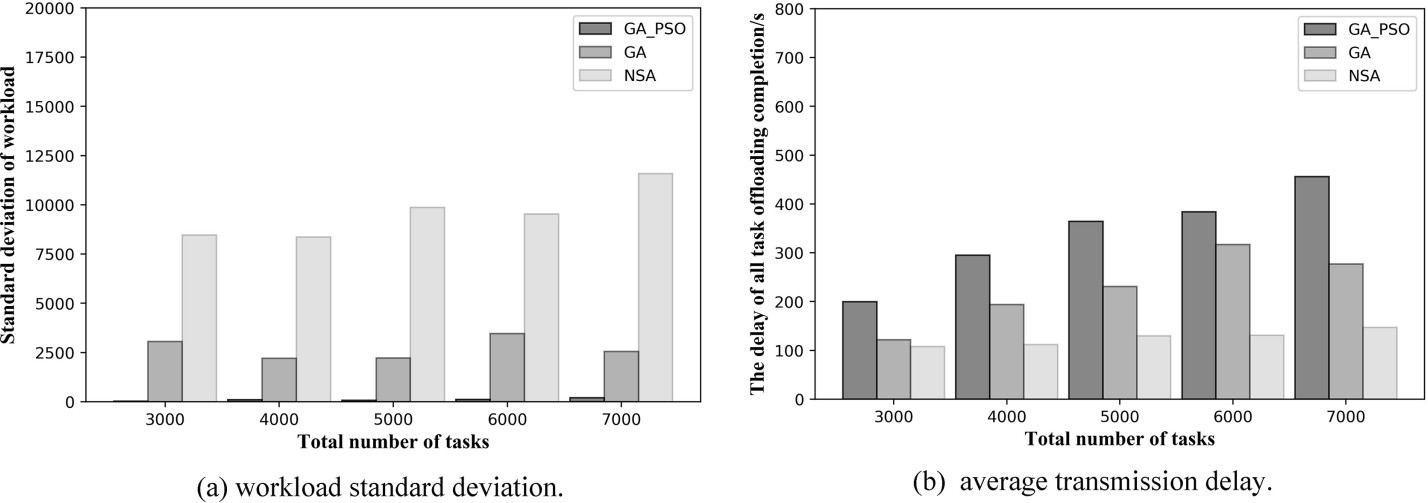

(a) workload standard deviation.

(b) average transmission delay.

**Figure 8** **(A) Workload standard deviation and (B) average transmission delay of various algorithm offloading strategies in network topology scenario 2.**

deviation value, the more balanced the workload among the edge servers. We conclude that in network scenario 1, the GA-PSO algorithm outperforms the GA and NSA algorithms in load balancing optimization. In terms of specific numerical performance, when the total number of tasks is different, the GA-PSO algorithm achieves values that are 92–96% lower than GA and 98% lower than NSA. At the same time, the values obtained by the GA algorithm are also 76–80% lower than those obtained by the NSA.

In Fig. 7B, in the topology of the first network scene, we can see that the offloading completion delay obtained by NSA algorithm is the lowest, while the delay value obtained by GA-PSO algorithm is the highest. This means that the NSA algorithm has the best performance in terms of latency optimization, followed by the GA algorithm and GA-PSO algorithm. In terms of specific numerical performance, when the total number of tasks is different, the values obtained by NSA algorithm are respectively 35–46% and 37–50% lower than those obtained by GA and GA-PSO algorithm, and the values obtained by GA algorithm are also 5–20% lower than those obtained by GA-PSO algorithm.

In the NSA algorithm, the server with the fewest hops is chosen to offload all the tasks received by the same AP. As a result, the tasks received by several APs without deployed edge servers are offloaded to one server for computation. At this point, the offloading completion delay for all tasks can be minimized, but this also results in the highest workload standard deviation values, making the workload of each edge server highly imbalanced. The GA algorithm selects the edge servers with less workload and offload hops for offloading, and achieves mediocre performance in terms of latency and workload. The GA-PSO algorithm refines the assignment of tasks. The tasks received by the same AP may be offloaded to different edge servers for computation, with different completion delays for each task. It minimizes the load imbalance among edge servers, but has higher latency than the other two algorithms.

Let us look at the resulting graphs in the second network topology scenario. Observing Fig. 8A, we can conclude that the GA-PSO algorithm also outperforms the GA and NSA algorithms for workload balancing in network scenario 2. The standard deviation of workload is different from that in scenario 1 in specific value. The GA-PSO algorithm yields a value that is 92–95% lower than GA and 95–98% lower than NSA. At the same time, the values obtained by the GA algorithm are also 63–73% lower than those obtained by the NSA. By looking at Fig. 8B, we can see that in network scenario 2, the NSA algorithm performs the best in delay optimization, followed by the GA algorithm and GA-PSO. In terms of specific numerical performance, when the total number of tasks is different, the values obtained by the NSA algorithm are 42–50% and 45–62% lower than those obtained by the GA and GA-PSO algorithms, respectively, and the values obtained by the GA algorithm are also 20–30% lower than those obtained by the GA-PSO algorithm.

A further comparison is made by the numerical results for the two scenarios above: while NSA can achieve the lowest offloading completion delay, which is 30–50% lower than GA and GA-PSO numerically, the standard deviation of the NSA workload is 70% and 90% higher than GA and GA-PSO, respectively. Hence, the effect of NSA optimization is relatively modest. Although the offloading completion delay of GA-PSO is 10–20% higher than that of GA, the standard deviation of workload of GA-PSO is 90% lower than that of GA. Therefore, GA-PSO has the best optimization effect.

In the same scenario, as the total number of tasks increases, the standard deviation of the workload obtained by the NSA algorithm also increases substantially. However, both GA and GA-PSO show some fluctuations as the total number of tasks increases, although the standard deviation of the overall workload increases. Moreover, the fluctuations in GA are more pronounced. Look closely at Figs. 7 and 8: in network topology scenario 1, when the total number of tasks increases from 4,000 to 5,000, the GA task offloading delay and the standard deviation of the workload decrease accordingly. In network topology scenario 2, when the total number of tasks increases from 6,000 to 7,000, the GA task offloading latency and the standard deviation of the workload also decrease. This is related to the GA algorithm design, where the tasks received by each AP are offloaded to the server represented in the gene. Network offloading decisions differ when the total number of tasks on the network increases. In this case, the tasks received by the same AP are offloaded to different edge servers. As a result, the offloading completion delay decreases for all tasks received by the same AP, and the workload of some servers will also modify, which will affect the standard deviation of the workload of all servers in the network.

At the same time, to validate the effectiveness of the proposed algorithm in more scenarios, we conduct experiments when the number of deployed edge servers is 4 and 6. Figures 9 and 10 show the comparison of the performance results of the three algorithms when the number of deployed edge servers is 4 and 6, respectively. In both Figs. 9A and 10A, we can see that GA-PSO has the lowest standard deviation of workload, while NSA has the highest standard deviation. GA-PSO is nicely optimized for load balancing. In Figs. 9B and 10B, we can see that NSA has the lowest task offloading completion delay, while

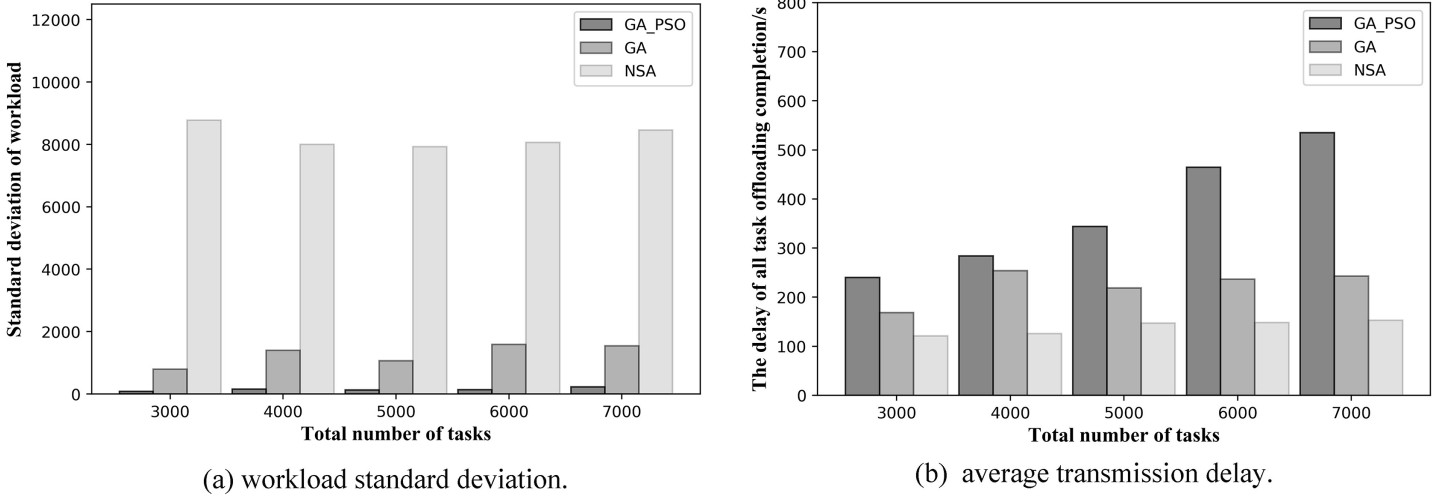

(a) workload standard deviation.  (b) average transmission delay.

**Figure 9 (A) Workload standard deviation and (B) average transmission delay of various algorithm offloading strategies in network topology scenario 3.**

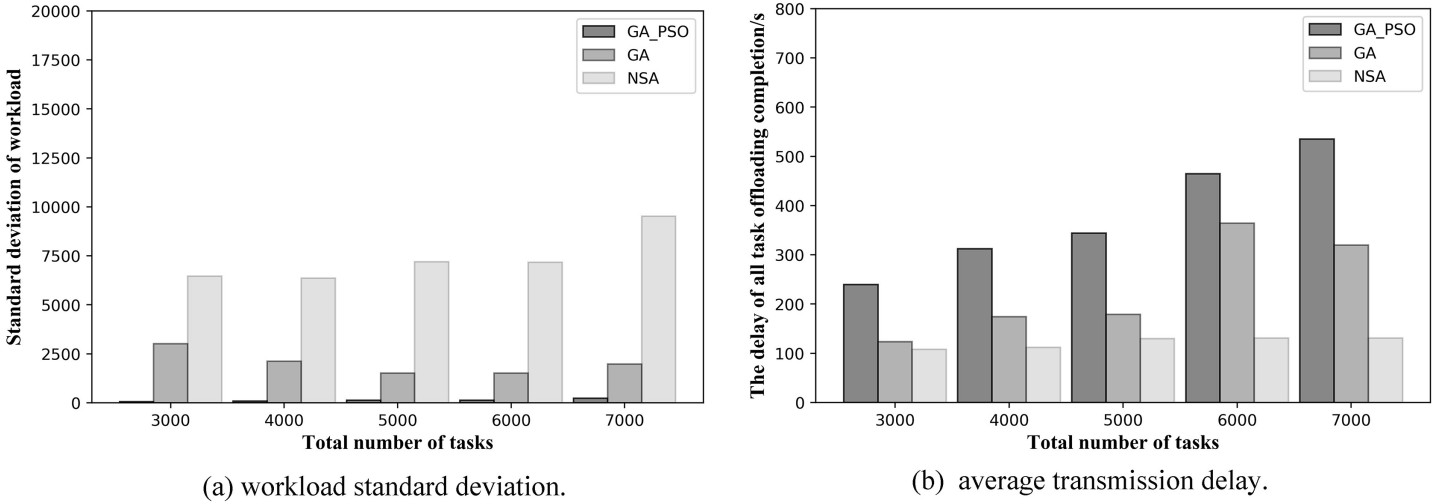

(a) workload standard deviation.  (b) average transmission delay.

**Figure 10 (A) Workload standard deviation and (B) average transmission delay of various algorithm offloading strategies in network topology scenario 4.**

GA-PSO algorithm has the higher task offloading completion delay. Therefore, our proposed GA-PSO is also effective when the number of edge servers is different.

We choose the fitness values when the total number of tasks is intermediate for comparison. Figure 11 shows the convergence results when the total number of tasks is 5,000 for the four network topological scenarios. As can be seen in Fig. 11, the algorithm is able to converge within 40 iterations in all four network topology scenarios, showing good convergence.

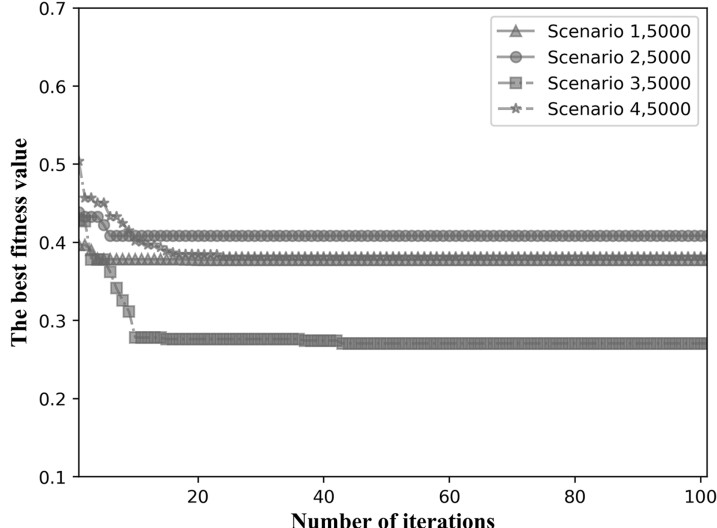

**Figure 11 Fitness values for the four network scenarios when the total number of tasks is 5,000.**

Based on the above observations, although the offloading delay of GA-PSO is slightly higher than that of GA and NSA, GA-PSO can achieve the lowest standard deviation of the workload for different network scenarios, thus balancing the load of edge servers.

## CONCLUSION

In this article, we address the multi-objective optimization problem of simultaneously optimizing server workload and offloading latency in networks with limited deployment of edge servers. We have presented a hybrid genetic particle swarm optimization algorithm based on two-dimensional genes. This method has the following two innovations: (1) In the algorithm, we consider the task received by AP to be offloaded by group. (2) When the task is offloaded in the algorithm, the selection of multi-hop path we adopt Dijkstra algorithm to select the shortest path between AP nodes. Experimental results show that PSO-GA can achieve the lowest standard deviation of workloads across different network topologies in multi-hop MEC systems with limited server resources, although it has average performance in terms of task offloading delay. At the same time, GA-PSO converges well. During the experiments, we found that the distribution of task data had an influential effect on the results. Therefore, we would like to further investigate and adopt different task offloading methods to optimize the edge server load and task offloading latency under different task distribution patterns.

### Funding

This work was supported by the fund from the Network and Data Security Key Laboratory of Sichuan Province, UESTC (NO. NDS2021-7), the Sichuan Province General Education Scientific Research (NO. 2019514), the Open Project of National Intelligent Society

Governance Testing Area (NO. ZNZL2023A04), the Research on Intelligent Access Control Technology for heterogeneous networks (CXHCL202201), the Meteorological Information and Signal Processing Key Laboratory of Sichuan Higher Education Institutes of Chengdu University of Information Technology, the fund of the Scientific and Technological Activities for Overseas Students of Sichuan Province 2022(30) and funds from the Sichuan Provincial Department of Human Resources and Social Welfare "Researches on Key Issues of Edge Computing Server Deployment and Computing task Offloading". The funders had no role in study design, data collection and analysis, decision to publish, or preparation of the manuscript.

### Grant Disclosures
The following grant information was disclosed by the authors:
Network and Data Security Key Laboratory of Sichuan Province, UESTC: NDS2021-7.
Sichuan Province General Education Scientific Research: 2019514.
Open Project of National Intelligent Society Governance Testing Area: ZNZL2023A04.
Research on Intelligent Access Control Technology: CXHCL202201.
Meteorological Information and Signal Processing Key Laboratory of Sichuan Higher Education Institutes of Chengdu University of Information Technology.
Scientific and Technological Activities for Overseas Students of Sichuan Province.
Sichuan Provincial Department of Human Resources and Social Welfare.

### Competing Interests
The authors declare that they have no competing interests.

### Author Contributions
- Wenzao Li conceived and designed the experiments, performed the computation work, prepared figures and/or tables, and approved the final draft.
- Xiulan Sun conceived and designed the experiments, performed the experiments, performed the computation work, prepared figures and/or tables, and approved the final draft.
- Bing Wan conceived and designed the experiments, analyzed the data, prepared figures and/or tables, authored or reviewed drafts of the article, and approved the final draft.
- Hantao Liu conceived and designed the experiments, analyzed the data, prepared figures and/or tables, authored or reviewed drafts of the article, and approved the final draft.
- Jie Fang performed the experiments, analyzed the data, performed the computation work, authored or reviewed drafts of the article, and approved the final draft.
- Zhan Wen performed the experiments, analyzed the data, performed the computation work, authored or reviewed drafts of the article, and approved the final draft.

### Data Availability
The code and raw data are available in the Supplemental Files.

## Supplemental Information

Supplemental information for this article can be found online at http://dx.doi.org/10.7717/peerj-cs.1273#supplemental-information.

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
