# Peer review of "A hybrid GA-PSO strategy for computing task offloading towards MES scenarios"

_PeerJ Computer Science, doi:10.7717/peerj-cs.1273_

## Round 0.1 · original submission · Major Revisions

In the content of the article, it is seen that a significant revision is necessary in the referee suggestions and the following issues. In this paper, uncertainties should be removed and desired corrections should be discussed in detail.
1) The authors compare these results with current research on the subject.
2) Experimental results are poorly handled and this section needs improvement.
3) The conclusion section should be rewritten to reflect the content of the article.

·

Basic reporting

The author must consider restructuring the entire manuscript. Especially the numbering of sections and sub-sections.

The study offers a solution The transmission delay and processing delay caused by the selection of AP and MES will affect the performance of MEC. However, the author needs to provide more background information on the following
1 Mobile Edge Servers (MES)
2. Mobile Edge Computing (MEC)

The related work section (line 294) should be removed and reintroduced just after introduction and background section.

Experimental design

The experimental design design used in the research is appropriate.

Validity of the findings

The findings of the study are valid: According to simulation results, the suggested hybrid genetic particle swarm technique based on two-dimensional particles performs well in terms of lowering the standard deviation of server workload.

The data provided supports the conclusion of the study

Additional comments

No comment

Reviewer 2 ·

Basic reporting

no comment

Experimental design

1.The experimental part lacks the comparison results with the latest algorithm, and algorithm advantages are insufficient.
2.The experimental part lacks an explanation of the experimental parameter setting, which will affect the credibility of the experimental results.
3.The simulation results of the manuscript are not detailed enough. I suggest the authors analyze the simulation results in-depth and provide more details.

Validity of the findings

1.The paper lacks a theoretical proof of the proposed algorithm.

Additional comments

Many scholars have conducted in-depth research on task offloading under resources are limited. authors should make relevant comparisons with these studies.

·

Basic reporting

This paper discusses the task offloading algorithm in the mobile edge computing environment, and establishes a system model according to the sparsity of MES deployment in real scenarios. The authors formulate a joint objective optimization problem considering the load balancing between MES and the propagation delay of multi-hop communication. And proposed an hybrid genetic particle swarm algorithm based on two-dimensional genes(GA-PSO) to solve it. Finally, the author compares the GA-PSO algorithm with NSA and GA through simulation experiments, and proves the superiority of the algorithm in balancing the workload. The research questions of this article are clearly defined, and the content is reasonable and novel but there are still some problems needs to modify.

First of all, there are grammatical issues. The English expressions of lines 40-41, 60-61, and 66 of the article should be improved. Similarly, the method of dividing tasks where located at the lines 183-184 need to be discribed more clearly.

secondly, In the System Model section, the content of figure 1 is too simple. The author could display part of the content in lines 104-122 on the figure to make the picture more detailed.

Experimental design

In paragraphs 199-207 of the article, the author introduced the dijkstra algorithm to solve the optimal transmission path problem, but there are other graph-based algorithms to solve the shortest path problem, such as the Floyd algorithm, A* algorithm, etc. Please further explain the reason for choosing dijkstra algorithm.

Validity of the findings

In the section of Performance Evaluation of Algorithm, it can be concluded from Figure 5 and Figure 6 that under the condition of same number of tasks, the algorithm which can achieve the best load balancing effect also endure the maximum latency. In other words, the latency optimization effect of GA-PSO, GA, NSA is gradually improved, but the workload of NSA are more unbalanced compare to the others. Please explain this phenomenon from the perspective of the underlying logic of the algorithm.

Figure 5 in the Performance Evaluation of Algorithm chapter shows that when the total number of tasks increases from 4000 to 5000, the GA task delay and the standard deviation of the workload decrease correspondingly. What is the reason for this phenomenon?

---

## Round 0.2 · accepted · Accept

The authors have satisfactorily addressed most of referees' concerns. So the paper is accepted for publication.

·

Basic reporting

The language used is easy to understand and conveys the correct message. I congratulate the author on the choice of literature references which has provided a clear background on the study. figures are easy to understand, the results are accurate which proves the efficacy of the methodology.

Experimental design

The research matches the scope of the journal. the research questions are relevant. the results fill up the research gap that were lacking in previous studies.

Validity of the findings

The conclusions are valid. the algorithms and formulas are robust, the same can be said of the findings

Additional comments

References are sound and recent.

·

Basic reporting

no comment.

Experimental design

no comment.

Validity of the findings

no comment.